# “To Be Treated as a Person and Not as a Disease Entity”—Expectations of People with Visual Impairments towards Primary Healthcare: Results of the Mixed-Method Survey in Poland

**DOI:** 10.3390/ijerph192013519

**Published:** 2022-10-19

**Authors:** Katarzyna Weronika Binder-Olibrowska, Maciek Godycki-Ćwirko, Magdalena Agnieszka Wrzesińska

**Affiliations:** 1Department of Psychosocial Rehabilitation, Faculty of Health Sciences, Medical University of Lodz, Lindleya 6, 90-131 Lodz, Poland; 2Centre for Family and Community Medicine, Faculty of Medical Sciences, Medical University of Lodz, Kopcinskiego 20, 90-153 Lodz, Poland

**Keywords:** disability, general practice, health services research, healthcare needs, patient expectations, patient preferences, Poland, primary health care, quality of care, visual impairment

## Abstract

Primary care is the core part of the Polish healthcare system. Improving its quality for vulnerable populations is among the principal goals of global and national health policies. Identifying patients’ needs is critical in this process. People who are blind or have low vision often demonstrate comorbidities and require more specific healthcare. The aim of this study was to explore the needs of Polish persons with visual impairments when they use primary care services. 219 respondents answered the “Patient value” questionnaire from the project Quality and Costs of Primary Care in Europe (QUALICOPC) and an open question regarding additional patients’ needs. Statistical and content analyses were used. The expectations of the study group regarding primary care appeared to be higher than those described in studies among other populations. Equity and accessibility were the most valued dimensions of care. Among particular aspects of care, those connected with psychosocial competencies and awareness of disability among medical staff appeared most frequently. Some personal characteristics were associated with preferences, including age, gender, longstanding conditions, quality of life, and disability-related variables. Our study indicates a need for multilevel interventions in legislation, economics, and medical staff training, with the people-centered approach as the option maximizing chances to meet diverse healthcare needs arising from particular disabilities.

## 1. Introduction

### 1.1. Primary Health Care in Poland

Primary health care (PHC) remains essential to meet the demand of “health for all” proclaimed in the Alma Ata Declaration adopted at the World Health Assembly in 1978 and renewed in Astana in 2018, and it is the first point of contact with health care systems for individuals, families, and communities [1]. The organization of PHC differs between countries. In Poland, PHC services are financed from public funds and comprise diagnostic, therapeutic, and preventive interventions which are provided by a primary care physician (PCP), nurse, and midwife chosen by the patient [2]. According to the European Health Interview Survey [3] published in 2019, about 81% of Polish patients used PHC. In 2020, between 45.2% and 62.4% of out-patient health care consultations (including also dental and specialized consultations) were realized by PHC in particular voivodeships (the highest-level administrative regions of Poland), also at nights and weekends [4]. From 2020 to 2022, at COVID-19 pandemic time, Polish PHC developed teleconsultations, which earlier were legally possible but rarely used [5].

### 1.2. Disabilities, including Visual Impairments, and Health Situation

World Health Organization (WHO) highlights the importance of the people-centered approach in PHC and the necessity of monitoring its quality and equity to find out if it meets health needs of various groups, including people with disabilities (PwDs) [6]. Additionally, the Polish healthcare system is obliged to ensure special health care for PwDs [7]. PwDs are characterized with poor overall health status in both less and highly developed countries [8,9] and, consequently, higher and specific health needs [8,10]. At the same time, they are at risk of attitudinal, physical, communication, and financial barriers, stigmatization, and discrimination when using health care [11]. There are also disparities regarding specific disabilities, e.g., spa hospitals and pharmacies are much better adapted for the needs of people with physical disabilities than for those with visual impairment (VI) [4]. For years people with visual impairments (PVIs) have been among those who face difficulties while obtaining medical care [12,13,14]. Their disability results in specific needs and is associated with a higher incidence of health ailments and chronic diseases as well as lower levels of well-being, self-rated health, and health-related quality of life [15,16,17,18,19,20] when comparing to the general population.

### 1.3. Patients’ Expectations—Earlier Studies and a Gap in PVIs Surveying

The awareness of patients’ needs and expectations improves the quality of health care [21]. Therefore, patients’ expectations and values have found among key issues of the international project Quality and Costs of Primary Care in Europe (QUALICOPC), coordinated by the Netherlands Institute for Health Systems Research (NIVEL), and conducted between 2010 and 2014. QUALICOPC aimed at assessing the quality, costs, and equity of primary care from patients’ and physicians’ perspectives. 34 countries, including Poland, took part in this project [22]. 219 patients, mostly females, aged 18 to 83 years participated in the part of the Polish study regarding patients’ expectations. The “Patient Values Questionnaire” (PVQ) was then validated as a tool for measurement of patients’ expectations towards PHC accessibility, continuity, equity, and quality [23,24].

Polish and foreign studies have analyzed the relationship between different patients’ characteristics, such as socio-demographic variables, self-rated health, the occurrence of chronic diseases, and their expectations [25,26,27,28]. However, the authors of such studies have focused less on a relationship between patients’ disabilities and their expectations regarding the quality of health care. We have not also found a study in which PVIs’ expectations would be examined with a similar tool as the one used for examining the general population. The last Polish survey of PVIs’ satisfaction with services provided by a family physician consisted of only four questions on few PCP’s tasks, regarding management of rehabilitation process and basic communication skills [29]. Nevertheless, an analysis of results revealed a need to improve communication skills of medical staff as they are considered an important part of coordinated support for PVIs.

### 1.4. Objectives

Considering the gap in research, the goal of the study is to:assess the general level of PVIs’ expectations towards the Accessibility, Continuity, Equity, and Quality of PHC;determine if PVIs’ sociodemographic, health, and disability characteristics are associated with the level of their expectations towards the abovementioned PHC dimensions;identify aspects of PHC considered as the most important by PVIs;compare expectations of poor-sighted and blind people towards PHC;establish expectations of PVIs related to the specificity of functioning with blindness and low vision in PHC.

## 2. Materials and Methods

### 2.1. Study Design

This cross-sectional questionnaire-based study included 222 persons, aged 18 years old or older, who self-declared having VI and signed an informed consent for voluntary participation in the study. Although few degrees of VI may be distinguished, for simplicity of description and out of the belief that its variation is too vast to reflect it appropriately, we divided the participants into two groups—respondents marked one of the options: I am a poor-sighted or blind person. VI is then an umbrella term for all of them. Due to the study design, we had to rely on self-declarations. In Poland, people are defined as poor-sighted or blind based on medical or legal definitions. According to the first one, contained in the section “Visual impairment including blindness” of The International Classification of Diseases (ICD-10), a blind person has no sense of light, achieves visual acuity with maximum spectacle correction of no more than 0.05, or has a visual field narrowed to no more than 20° in radius [30]. According to the Polish legal definition, a blind person has a visual acuity of 0 to 5/50, and a visual field narrowed to 30 degrees. On the other hand, a poor-sighted person is one whose visual acuity is between 0.05 and 0.3 and whose visual field may be limited to 30 degrees. From the functional perspective, PVIs have difficulties performing daily activities, despite the best available lens correction [31]. The exclusion criteria were absence of any of these conditions or difficulty understanding the questions. Due to a significant lack of answers, three questionnaires were excluded from the analyses.

Accidental selection and snowball sampling were used to engage as many participants as possible into the study. Information about the project had been disseminated via Internet forums, schools, sport clubs, ophthalmology clinics and departments, and associations for PVIs. Data were collected between May and July 2021 via Google Form, e-mail, paper, or phone calls conducted by a trained interviewer. Paper-pencil questionnaires were prepared in an enlarged font and paper leaflets included a QR code to read the information electronically. The Braille questionnaire had not been used by anyone.

Respondents took part in this survey anonymously and voluntarily. The study was conducted in accordance with the guidelines of the Declaration of Helsinki after reviewing the study protocol by the Bioethics Committee of the Medical University of Lodz (no RNN/114/21IKE).

### 2.2. Measures

The authors used the Polish version of the PVQ. In its original version [23,32], the questionnaire consisted of 19 questions, 12 of which concerned characteristics of the participants (Appendix A), whereas seven with sub-items referred to a total of 47 PHC variables in terms of their importance to the respondent. In the part of PVQ concerning PHC, the participants responded to each item by checking one of four options on an ordinal scale: not important, rather not important, important, and very important, for which in the result analysis the following values: −1, −0.5, 0.5 and 1 were used, respectively. The examined variables in this section covered the following dimensions of care: its Accessibility, Continuity, Equity, and Quality of care. The last one consisted of four aspects: socio-emotional behaviors, preparation for consultation, course of the visit, and post-consultation management. In the study among general population of Poland the α-Cronbach coefficient calculated for the three domains of quality of care demonstrated the following values: Accessibility of care α = 0.79; Continuity of care α = 0.64; Quality of service α = 0.93. The Equity dimension comprised a single question, which did not allow to determine the Cronbach’s α coefficient for it.

PVQ was modified for the purposes of current research regarding the specific functioning of PVIs. Firstly, a pilot study was conducted in the target population consisting of ten people in order to check its comprehensibility by the subjects, and then some changes in the content of the questions were made. In the part concerning respondents characteristics some items were omitted and some additional occurred. In the questions regarding patients’ values one change concerned a strictly linguistic issue and others were related to the disability. All changes with their rationale have been presented in Appendix A. To compare the results obtained with those of the nationwide survey, in our study we kept the division into four dimensions of care and the adopted principles of assessing the scope of expectations [23,24]. We assessed also the reliability of the modified PVQ using the α-Cronbach coefficient, and it was as follows: Accessibility α = 0.62, Continuity α = 0.48, and Quality of care α = 0.90.

To learn more about PVIs expectations and needs when using PHC, respondents were also asked an open-ended question: *“What else is important to a poor-sighted or blind person before, during and after the visit to PCP?”*

### 2.3. Data Analyses

Descriptive statistics were used in the characteristics of respondents. To compare the frequency of each trait variation in the study groups, the chi2 independence test and Fisher’s exact test were used. Before comparing the means in the studied groups and subgroups, the conformity of the distributions of individual variables to a normal distribution was checked using the Shapiro–Wilk normality test. As the distributions of the analyzed variables regarding expectations for Availability, Continuity, Quality, and Equity of Care differed significantly from the normal distribution, nonparametric tests were used to compare means. For a comparison of means in two independent subgroups, the Mann–Whitney test was used, and for a comparison of several subgroups, the nonparametric Kruskal–Wallis test was used. To assess the size of the effect, the coefficient Φ, d Cohen’s and Epsilon-squared were calculated. Odds ratios were not calculated because the conditions of their applicability were not met: some responses in PVQ were not chosen by any participant [33]. The reliability of PVQ was measured with the α-Cronbach coefficient. *p* < 0.05 was considered statistically significant.

Respondents’ answers to the open-ended question were elaborated with the content analysis [34]. All replies were recorded in a computer database. After preparing the computer database, two coders (K.W.B.-O. and M.A.W.), initially separately and then together, reviewed the responses and proposed a framework of codes covering the subject area. After the discussion and determining common frame, codes were manually matched to respondents’ statements and rediscussed. The number and percentage distribution of people who gave answers assigned to each category were provided. The poor-sighted and blind respondents’ answers were compared using chi2 and *p*-value.

Statistical analyses were conducted using Statistica software v. 13.3 (StatSoft, Kraków, Poland, www.stasoft.pl; accessed on 1 September 2021).

## 3. Results

### 3.1. Sample Socio-Demographic, Disability, and Health Characteristics

One hundred and twenty-six women and 93 men took part in the study. The mean age of respondents was 50.74; SD ± 17.16. Most of the participants (*n* = 119; 54.3%) lived in cities with a population of more than 100 k, 31.5% (*n* = 69) of them were inhabitants of the urban areas between 50 and 100 k, and over 14% (*n* = 31) persons lived in rural areas. The majority of participants have completed their secondary education (44.5%; *n* = 98) and primary education was the least common (16.9%; *n* = 37). A similar percentage of single and married people was observed (37.9%; *n* = 83); the rest of the respondents were divorced or widowed. More than a half of the surveyed participants were pensioners, more than 40% were employed, and every four respondent was retired. 11.1% of the blind and one poor-sighted person were self-employed (chi2 = 11.161; *p* < 0.001). Half of the respondents (*n* = 109) described their income as average and over 40% (*n* = 89) claimed it was lower than average. For about half of the study group (*n* = 103), the income they earned was the main source of their household income.

The study group consisted of 156 poor-sighted and 63 blind individuals. The male group was dominated by blind people and the female group by partially sighted. More than 40% of respondents had also another disability; more often partially sighted than blind people: 44.9% vs. 27.0%, chi2 = 5.997; *p* < 0.05. Forty percent of the poor-sighted and only one blind person declared moving independently (chi2 = 31.394, *p* = 0.000). In turn, the blind more often than the poor-sighted used the service of a PwDs’ assistant (46.1% vs. 29.5; chi2 = 10.314; *p* < 0.01); in the whole group, it was about one third of all respondents.

Both partially sighted and blind people most often rated their health status as satisfactory (43.6% and 39.7%) or good (35.9% and 34.9%). Almost two-thirds of all participants reported having a chronic disease. Its diagnosis was significantly more common among the poor-sighted than in the blind (58.6% vs. 50.8%; chi2 = 6.130; *p* < 0.05), and in poor-sighted women than in men (74.5% vs. 58.6%; chi2 = 4.259; *p* < 0.05). The quality of life in the last four weeks was assessed as good by more than half of the total number of respondents. The blind were significantly more likely (chi2 = 12.400; *p* < 0.05) to describe it as very good when compared to the poor-sighted (22.2% vs. 9.0%), who in turn claimed that it was “neither good nor bad” (34.0% vs. 15.9%).

The majority of respondents (82.2%) had used PHC services within six months before the survey. Most poor-sighted participants (34%) attended three to five visits during that time, whereas more than 25% of the blind did not use PHC services and nearly ¼ did it twice (*p* < 0.05) (Appendix A).

### 3.2. General Levels of PVIs Expectations towards the Main Dimensions of PHC

Equity and Accessibility appeared to be the most important dimensions of PHC for respondents, representing the highest mean values (0.70 and 0.69, respectively). The Continuity dimension, with the mean value of 0.43, was the least valued by participants (Figure 1). Poor-sighted and blind respondents did not differ significantly in their level of expectations in particular dimensions (*p* > 0.05).

### 3.3. PVIs Characteristics and Expectations towards Four PHC Dimensions

In the whole surveyed group, the level of expectations in women was significantly higher than in men (*p* < 0.05) with regard to the dimensions of the Continuity and Equity. Among poor-sighted respondents, women demonstrated a higher level of expectations than men in all surveyed dimensions—in relation to the Continuity and the Quality of Service *p* < 0.01, in relation to the Accessibility and Equity *p* < 0.05. Only expectations for the Continuity of care increased with the age of the respondents (*p* < 0.001). The place of residence, education, and health status did not contribute to respondents’ different expectations regarding the four dimensions of care discussed above. There were also no differences in the levels of expectations depending on the level of VI or whether the respondent had one or more disabilities (*p* > 0.05). Only the Continuity dimension was significantly more important for participants with chronic diseases than for those without (*p* < 0.001). In the entire group, higher quality of life was associated with a significantly higher level of expectations regarding the Accessibility of PHC (*p* < 0.05). Additionally, respondents who moved around with the help of others or assistive tools revealed higher expectations for Accessibility than those moving independently (*p* < 0.05). According to the Quality dimension, the highest level of expectations was found among participants who most often used the support of PwDs’ assistant (*p* < 0.05) (Table 1).

### 3.4. Hierarchy of PHC Values including the Level of Respondents’ VI

Table 2 shows the percentages of respondents who rated each PVQ item as *not important, rather not important, important*, or *very important* with the division to the four PHC dimensions. Moreover, it presents the percentage of *very important* answers among blind and poor-sighted participants. All respondents assessed PCP knowing when to refer, listening attentively, and understanding the patient as important or very important. Understanding PCP’s explanations and easy getting appointments were assessed similarly. The least number of respondents expected that psychosocial problems can be discussed during consultation and that PCP knows the patient’s living situation.

Respondents differed in some preferences depending on the level of their VI. More blind persons than those poor-sighted expected PCP to provide the patient with trusted sources of information (*p* < 0.01) and to be interested in other problems of the patient (*p* < 0.05). Listening carefully by a PCP was more significant for the blind than for the second group (*p* < 0.01), and it was the most important value for them during the consultation, whereas the poor-sighted in the first place checked reasonableness of the referral to specialist.

### 3.5. PVIs Expectations towards PHC—Voice of Respondents

Over 70% (*n* = 155) of participants answered the question *“What else is important for a poor-sighted or blind person before, during, and after the visit to PHC?”* The blind participants answered significantly more often than the poor-sighted (*n* = 54, 85.7% vs. *n* = 101, 64.7%; chi2 = 9.550; *p* < 0.01). The frequency of giving more than one answer did not differ significantly between the groups (*n* = 19, 35.2% of the blind respondents and *n* = 23, 22.8% of the poor sighted; chi2 = 2.754; *p* > 0.05).

A content analysis of the responses enabled to obtain four categories describing PVIs’ needs and expectations. These were as follows: psychosocial competencies of the staff (56.8% answers in the total group); primary care clinic available to PwDs (44.5%); help adequate to personal needs (28.4%), and PwDs-friendly medical procedures (26.4%) (Figure 2).

The most frequently indicated—psychosocial competencies of staff were the most relevant for 61.1% of the blind and 54.4% of the poor-sighted (chi2 = 0.634; *p* > 0.05). This category was mainly associated with communication skills, such as providing clear information, confirming that the patient has understood the message, asking about the patient’s needs, and proper non-verbal communication. Moreover, PVIs highlighted the importance of PCP knowledge regarding the visual disability and empathetic, kind, serious, and respectful attitude towards patients. Table 3 contains few distinguished subcategories with representative quotes. The following sentence summarizes it: “For me, it is important to be treated as a person and not as a disease entity.”

Responses from a significant percentage of participants further regarded accessibility in its broad sense. This aspect was significantly more valued by the poor-sighted than by the blind (53.5% vs. 27.8%; chi2 = 9.375; *p* < 0.01). To the greatest extent, this was related to removal of architectural barriers and adaptation of clinic space respecting needs of PVIs. Second aspect was the usage of equipment and new technologies (Table 4).

Moreover, one respondent described his recommendations for ensuring the quality of care in terms of accessibility:


*“I recommend more cooperation with nongovernmental organizations working for PwDs, audits of architectural, communication and information accessibility. In addition to staff training and mystery clients with disabilities, social campaigns should be organized and employees of PHC should be controlled suddenly and unannounced.”*


Ensuring accessibility is important for PVIs’ psychological well-being because it allows them to be independent and self-reliant:


*“For me, independence is very important and I don’t like the confusion that an appearance of a person with a white cane causes. I would prefer to be independent of the lady at the front desk, reception desk or security when I need to find the office or reception desk.”*


Regarding the help adequate to personal needs, respondents highlighted that medical staff should answer specific needs of patients with VIs (Table 5), and that sometimes it is essential to just offer help or ask what the PVIs needs. The third category was indicated slightly more often by the blind then by poor sighted (31.5% vs. 26.7%; chi2 = 0.391; *p* > 0.05).

As one statement shows, knowing of and responding to the patient’s needs is reflected in patient satisfaction:


*“I am satisfied with my primary care clinic—the people working there know me, they know about my disability, they help me in organizational matters, e.g., they drive me to the office, they watch my queue, they call me a cab after my visit.”*


Slightly more poor-sighted participants indicated the last category when compared to the blind (27.7% vs. 24.1%; chi2 = 0.240; *p* > 0.05). PwDs-friendly medical procedures, included responses related to the efficiency and organization of treatment as well as procedures that facilitate reception of medical services by PVIs. Respondents suggested, *inter alia*, that the doctor should:


*“inform patients what preventive examinations should be done, at what age and what is the closest locations for such examinations” and “take care of the same patients, because he knows them and their situation.”*


For the patients with low vision or blindness, it is also important *“that the doctor take eye conditions into account while planning the treatment (to identify potential side effects of medications or difficulties in performing certain activities by a person with visual impairment).”* One woman made a comment on having difficulty in taking medications: *“If the form of administered medication is cumbersome (e.g., a precise amount of medication or a particular number of drops should be placed onto a small spoon or into a syringe), the patient with visual impairment should be assisted by their clinic to take the medication. This also applies if the patient is a young child of a blind or poor-sighted person!”*

Moreover, a person with VI needs longer visit time and explanations to find PHC space:


*“During the visit [it is important] that the doctor does not rush you, informs you exactly what is going to be done and how it is going to be done, e.g., examination, gives you the opportunity to feel the examination area, e.g., the place of the recliner and does not rush you to do it. And after the visit, the patient should be able to contact the doctor by phone or in person at any time during the treatment course.”*


In addition, respondents considered the following items important for PVIs: possibility to visit clinics also on Saturdays, easier access to counseling at the patient’s home, and punctuality of visits. One respondent justified this need as follows: *“Sticking to a given appointment time [is important] because the blind person’s guide doesn’t usually have time or spend his or her valuable time.”* Moreover, PVIs would appreciate being allowed to jump the queue and personnel of medical centers not minding bringing guide dogs into the office by patients.

Furthermore, several people pointed out benefits of e-health solutions: a possibility of remote contact with the doctor, ordering prescriptions for ongoing medication by phone, online registration and online filling documentation, and medication dosage instructions available in the digital form, e.g., sent via e-mail by the PCP. Respondents explained it as follows:


*“It is also more difficult to weed out all sorts of questionnaires, statements, and documents on the spot. Thus, it is important that they be available in the electronic version and can be filled out before the visit.”*



*“The doctor who has written out documents to ZUS [Zakład Ubezpieczeń Społecznych, The Social Insurance Institution] should send them by e-mail so that the person with disabilities does not have to visit many specialists (e.g., ophthalmologist, neurologist, orthopedist, and finally ZUS).”*


Finally, a statement from one participant indicates that the needs described go beyond the issue addressed in this study.


*“I believe that it is highly important to design procedures which would make it easier for blind people to receive treatment at the primary care level, use specialized care, rehabilitation and sanatoriums.”*


## 4. Discussion

Although high-quality health care for PwDs is one of goals of international and national legislations [7,35,36], diverse needs arising from specific disabilities, including VI still have not been given consistent consideration [37,38]. Since taking into account the diverse needs of PwDs and investigation of their views increase the quality of health care [35], we aimed to enhance knowledge about expectations of Polish PVIs using primary care services, including their associations with different patients’ characteristics.

### 4.1. The Highest and the Lowest Valued PHC Dimensions by PVIs

Of the four PHC dimensions distinguished in Polish QUALICOPC and the current study, Equity and Accessibility were the most important for respondents in both surveys [23], and there were no statistical differences in any of dimensions according to the level of VI. However, PVIs demonstrated higher expectations in all measured dimensions of care than subjects from the general Polish population (Figure 3). It may be due to their specific needs and barriers they must break to access healthcare services and their willingness to be actively engaged in medical treatment [39]. As respondents clarified in answers to the open-ended question, they want to be treated equally and seriously, and easy access to healthcare services gives them a feeling of safety, especially when they are dependent on others or assistive devices.

Continuity of care was the least valued dimension in our and previous Polish survey [23]. In the current study, it was mainly because PVIs did not expect PCP to know their psychosocial situation. However, answering to open question, respondents highlighted that continuity of care is significant because it allows the staff to recognize their specific needs. Moreover, this dimension gained the lowest level of reliability measured with Cronbach’s alpha. Thus, it seems that the perspective of PwDs on what continuity of care means may differ from that of the general population. Future qualitative research will allow us to define it.

### 4.2. PVIs Characteristics Associated with Their Expectations towards PHC Dimensions

Literature review presents various patients’ characteristics which influence their expectations, which has been partly confirmed in our study. Of sociodemographic variables, only gender and age differed expectations of the study participants. A higher level of expectations observed in the female group corresponds to some previous results [25,40,41]. This could be explained by the fact that women are diagnosed with chronic diseases more frequently, are more prone to develop multimorbidity [42,43], visit physicians more often [3,44,45], and are more actively involved in consultations, i.e., by asking more questions than male patients [45]. Additionally, in our study, women reported to be affected by longstanding conditions more often than men. In consequence, they may be more focused on their health, more aware of their healthcare needs, and demonstrate a higher level of expectations. Moreover, poor-sighted women presented higher levels of expectations for all PHC dimensions comparing to poor-sighted men. With regard to the blind, women differed from men only with Continuity and Equity dimensions. Further research is necessary to identify the reason why there were more differences between poor-sighted male and females than those observed in the blind group.

Older age respondents demonstrated higher expectations regarding the Continuity of care dimension, which corresponds to previous Polish [23] and foreign [46,47] studies. Older patients more frequently experience multimorbidity [17,48,49], visit their PCPs more often [3] and are reluctant to change their PCP [50]. They may prefer to wait longer for a visit but be consulted by a familiar PCP [51]. Once the PCP is familiar with their medical and life background, patients do not have to repeat their story from the beginning, the relationship with the PCP becomes stronger [47], and the patient’s motivation to adhere to doctor’s orders may be higher [52,53]. A high rotation of providers and staff was ticked as one of barriers disrupting care coordination and continuity of care for PwDs, which resulted in choosing a different healthcare service provider, disturbances in relationships with known providers and the necessity to synthesize comprehensive information from different providers [54]. Continuity of care was seen by PwDs as a facilitator to access healthcare [55]. In the studies on the general population, continuity of care was associated with patients’ satisfaction [56], positive assessment of quality of care [50], and lower mortality [57,58].

A subjective assessment of the health status was insignificant for expectations in the PVIs group, which contrasted with the previous Polish study [23]. However, longstanding conditions were associated with higher expectations regarding the Continuity of care dimension in ours as well as some other studies [23,38]. An explanation for this observation can be similar to the one between age and the Continuity of care dimension. 

Respondents with VIs who regarded their quality of life as very good presented higher expectations for the Accessibility of care dimension. Similarly, in a Polish study conducted on patients with some chronic diseases [59], a lower quality of life corresponded with a lack of expectations for a physician or nurse. We assume that people who asses their quality of life as good may more often promote active lifestyle, be determined to care about their health and hence, appreciate access to healthcare more. In contrast, lower quality of life may be associated with worse health status, lack of motivation, and hope for health improvement.

Being poor-sighted or blind and affected by one or more disabilities did not contribute to any differences in participants’ expectations towards four PHC dimensions. However, two significant disability-related determinants for PVIs expectations were revealed. Moving outside with another person or assistive devices corresponded to higher expectations for PHC Accessibility, and using the help of a PwDs assistant was related to the Quality of care dimension. Probably, people who are provided with help experience more difficulties but also are more aware of possible obstacles and facilities they can meet. Besides, they have experience in cooperation with others and clarifying their needs.

### 4.3. The Most Valuable Aspects of PHC for PVIs Measured with PVQ

Like in other QUALICOPC studies [24,40,60], also in the present one, various communication skills of the medical staff appeared to be the most important value among the top ten. The interpersonal aspect of care was previously described as the one which mostly affected patients’ satisfaction [61]. Besides, similarly to Greek [60], Nordic [62] and previous Polish [24] surveyed, PVIs expected from the PCP clear instructions what to do in case their health deteriorates. However, in contrast to the Nordic study, it was the only one value among top-ten, associated with patients’ involvement indicated by the total study group. A PCP who knows when to refer the patient as well as the easy procedure of making an appointment were common for current and Icelandic respondents [62]. Another Accessibility value—answering the telephone within no time, appeared to be one of the top-ten variables only in current study.

Moreover, respondents from our study more frequently assessed particular items as very important than participants of other studies based on PVQ [23,40,60,62]. This may indicate a tendency of PwDs to value various aspects of healthcare higher than people without disabilities do. We hypothesize they can be more often aware stakeholders of medical and other services and also through trainings in self-advocacy they can be more familiar with their needs. On the other hand, since other studies were conducted about a decade before ours, levels of expectations could have generally increased among patients. Hence, this determines the updating of data in this area from international perspectives.

The study group demonstrated some differences regarding the hierarchy of values of the Quality of care dimension. A PCP who listens to the patient attentively, asks about other patient’s problems and informs them about reliable sources of information was more important for the blind than for the poor-sighted. It may be due to the inability to visually inspect the situation caused by blindness and the more severe impact of this disability on daily functioning, including information seeking, than that of the poor-sighted. Further qualitative research could provide more information about different preferences in patients with a various degree of vision impairment.

### 4.4. PVIs Voice about Their Needs When Using PHC

A content analysis of open question responses enabled us to identify four additional areas of PVIs’ needs when using PHC services: psychosocial competencies of staff, primary care clinic accessible for PwDs, help adequate to their personal needs, and PwDs-friendly medical procedures (listed by frequency of appearance). In summary, respondents highlighted the need to be treated as competent patients who would like the PCP to communicate directly with them, not their accompanies, and to focus on their current problem instead of disability. Psychosocial competencies of medical staff, including communication skills, empathy, disability, and eye condition awareness, were indicators of high-quality care. The vital role of nonverbal communication in patient-doctor relationships was emphasized. Accessibility was identified as the well-designed infrastructure of a PHC clinic, the help provided by medical staff or assistants, and solutions that make PVIs independent, including time for orientation in space, technologies, and e-health solutions. All mentioned needs have been described several times before [13,39,63,64,65,66,67,68]. However, they still need consideration because of existing barriers and changes regarding, inter alia, developing telemedicine [69,70].

Respondents’ statements in which they claim they would like to enter a PCP’s office without waiting in queues and going over the clinic with a guide dog, may imply that some people are unaware that they can already enjoy such a right, that rules are not always followed or that the group of people entitled to it is not sufficient.

In light of the open question analysis, the poor-sighted put significantly more attention on the accessibility of PHC clinics than the blind. This can be explained by the fact that the blind are more likely to use the help of other people and devices, while the poor-sighted rely on themselves. For the blind, on the other hand, high staff competencies are particularly important, as the medical personnel prepared to work with them is in itself a factor in facilitating accessibility. Moreover, the blind respondents were significantly more likely to state their opinions in an open question. This may be connected with their higher awareness of individual needs and laws and experience in expressing them. Again, further research is needed to determine the differences between people with various disability degrees.

### 4.5. Strengths and Limitations

To the best of our knowledge, the present study is the first one in which a standardized tool, the QUALICOPC questionnaire “Patient values”, has been applied to a group of PwDs. This is also the first study in which PVIs values were explored with the mixed method, including a quantitative and content analysis of data. Nevertheless, it is not free from certain limitations. A comparative analysis of results of the study group and those of other studies, based on PVQ, should be made with caution. In particular studies different PHC dimensions were distinguished [23,24,40,44,60,62]. Moreover, our study was conducted about a decade after the earlier QUALICOPC studies. We can assume that expectations are not much prone to changes but we cannot exclude it, since during this time some modifications in Polish PHC policy have been made and the study was implemented in COVID-19 time when a new idea of health policy had been forced. Due to numerous pandemic restrictions, common teleconsultations, and a lack of a register of PVIs in Polish primary care clinics, a direct contact with subjects visiting their PCP, as it was in studies conducted under the QUALICOPC project, was hardly possible. However, the number of participants is very similar to those from other QUALICOPC Patient-values studies [23,40,44,60]. Additionally, by varied methods of recruitment and data collection, as described in the Measures section, we did our best to gather opinions from a heterogenous group reflecting reality of the target population. On the other hand, indirect examination prevented us from achieving the participants’ certificates of disability, and only self-declarations, which ensured data protection and anonymity, were obtained.

The percentage and the length of the answers to the open question confirm that the addressed subject matter is noteworthy. Open-ended questions are valuable in enhancing the quality of healthcare [71,72], and mixed methods, which enable respondents to express their opinions, play an essential role in recognizing patients’ needs and measuring their satisfaction [73].

We are aware that neither quantitative data from PVQ nor answers to an open question can be generalized. Yet, we believe that such a mixed method was adequate to achieve the descriptive aim of this study. A content analysis confirmed special needs of PVIs. We believe that our study shows that known tools can be used in vulnerable populations after their adaptation. However, we might not obtain crucial information if we rely only on them.

## 5. Conclusions and Implications for Practice

Our study reveals that PVIs have in general the same needs and expectations as patients without disabilities—they want to be involved in the treatment, and appreciate good communication with medical staff. Depending on some personal characteristics, the expectations might differ, however. On the other hand, disabilities generate some additional needs regarding equity, continuity, accessibility, and quality of care. The quality of medical services provided to PwDs is affected by the ability to recognize these specific needs. This becomes possible via appropriate communication and information management with the use of knowledge, skills and experiences of both staff and patients [74]. To do this, multi- and interdisciplinary actions at different levels are needed. The first step includes the implementation and evaluation of standards and regulations [75]. These processes will allow one to constantly monitor healthcare providers and oblige them to respect the needs of vulnerable groups of patients. The second, economic aspect is related to expenditure which should be planned reasonably in order to eliminate architectural, information, and financial barriers that PwDs encounter. The economic aspect also involves ensuring proper financial management by healthcare providers. The third one is connected with education of medical staff and aims at raising disability-related awareness, improving skills and changing attitudes [9,64,76,77,78,79,80]. Last but not least, PVIs, who are the best source of knowledge about their needs, can participate in PHC quality auditing, but also should be prepared how to communicate their expectations, what their patient’s rights are, and how to take responsibility for their health. In addition, employing PwDs as staff members more frequently could help them to get rid of stigma and, in consequence, could improve their situation [79]. Such a multilevel approach is necessary because focusing only on some aspects does not yet guarantee equitable care [81].

Some PVIs’ needs, especially those arising from the disability specifics, are to a great extent those which were described previously. In turn, associations between patients’ characteristics and their expectations towards healthcare are inconsistent in several studies [40,82], and a literature review does not allow us to make unambiguous conclusions on them. As there is no general pattern of patients’ preferences, they should be regularly identified in the direct contact with medical staff members [83]. From this point of view, our results may constitute a call for action to develop patient or person-centered approach in Polish healthcare for PVIs. Bearing in mind the fact that we have such a huge diversity of expectations and needs, it seems to be the best way to provide high quality healthcare [76,84,85].

Moreover, the Polish healthcare system, which previously focused on homogenous patients in terms of their language and culture, faces nowadays a new challenge—provision of the care for refugees from Ukraine [86]. These nationals may face physical and mental health disturbances and face barriers regarding access to healthcare [87]. Therefore, refugees, including those with a disability, who are a vulnerable group at risk of unmet healthcare needs, require special attention and care [88].

## Figures and Tables

**Figure 1 ijerph-19-13519-f001:**
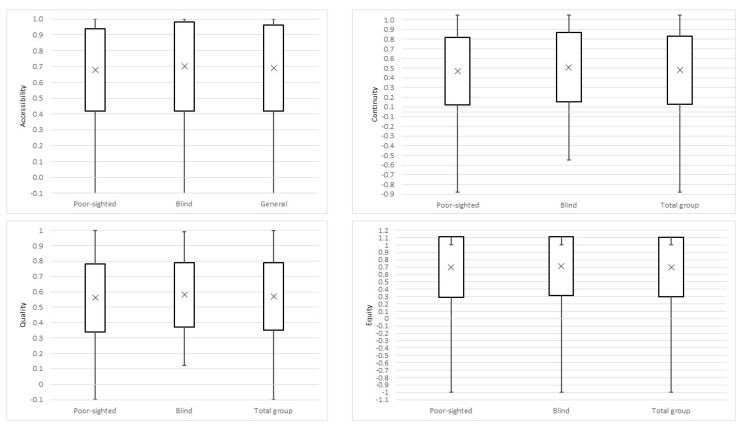
Mean PVIs’ expectations towards four PHC dimensions.

**Figure 2 ijerph-19-13519-f002:**
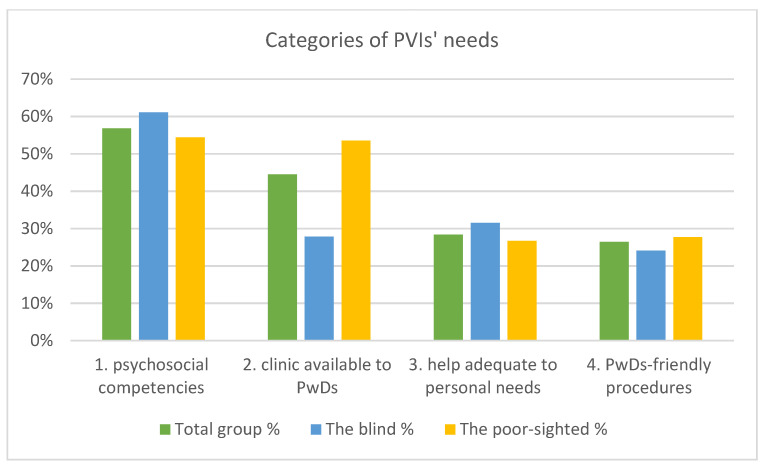
Categories of PVI’s needs in the total group and among the blind and poor-sighted.

**Figure 3 ijerph-19-13519-f003:**
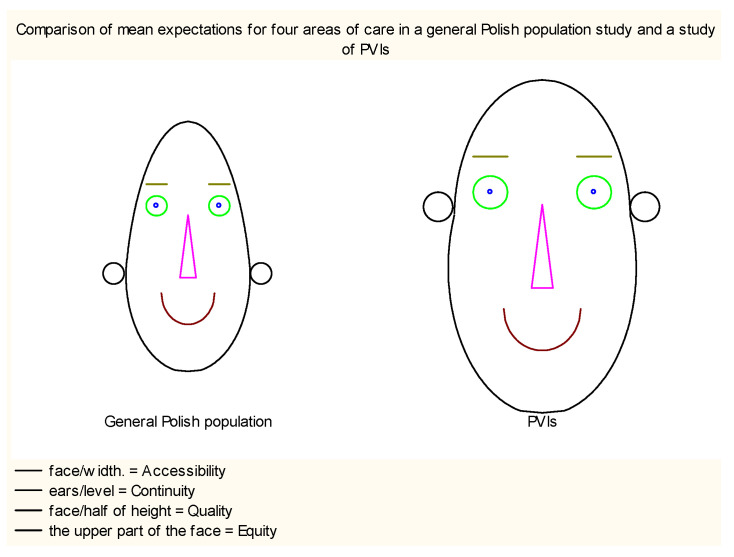
Visualization of expectations towards PHC in PVIs and the general population.

**Table 1 ijerph-19-13519-t001:** Expectations towards PHC according to sociodemographic, health-related and disability variables among PVIs.

	Accessibility		Continuity		Quality of Care		Equity	
	Mean	SD	Mean	SD	Mean	SD	Mean	SD
**Overall**	0.69	0.27	0.43	0.35	0.57	0.22	0.70	0.40
**Gender**								
Female	0.72	0.22	0.48	0.33	0.59	0.19	0.75	0.37
Male	0.64	0.31	0.37	0.37 ^a^	0.53	0.24	0.63	0.44 ^b^
**Gender in the poor-sighted group**								
Female	0.73	0.21	0.48	0.35	0.60	0.20	0.74	0.37
Male	0.61	0.32 ^c^	0.32	0.32 ^d^	0.50	0.25 ^e^	0.61	0.45 ^f^
**Gender in the blind group**								
Female	0.70	0.25	0.46	0.26	0.57	0.17	0.77	0.35
Male	0.70	0.30	0.46	0.42	0.59	0.24	0.66	0.43
**Age**								
18–39	0.69	0.26	0.31	0.37	0.56	0.22	0.65	0.43
40–59	0.66	0.28	0.41	0.33	0.53	0.22	0.75	0.35
60 and more	0.72	0.25	0.55	0.31 ^g^	0.61	0.21	0.70	0.35
**Place of residence**								
Rural	0.68	0.28	0.38	0.34	0.60	0.20	0.73	0.40
Urban to 50 k of inhabitants	0.68	0.26	0.44	0.36	0.53	0.22	0.68	0.40
Urban from 50 k to 100 k of inhabitants	0.58	0.32	0.36	0.35	0.55	0.21	0.67	0.35
Urban from 100 k to 500 k of inhabitants	0.68	0.28	0.47	0.33	0.58	0.21	0.72	0.31
Urban over 500 k of inhabitants	0.74	0.23	0.45	0.36	0.58	0.23	0.70	0.48
**Education level**								
Primary	0.78	0.21	0.47	0.34	0.61	0.19	0.69	0.38
Secondary	0.68	0.25	0.47	0.32	0.58	0.22	0.71	0.39
Higher	0.65	0.30	0.38	0.38	0.54	0.22	0.69	0.43
**Level of disability**								
Poor-sighted	0.68	0.26	0.42	0.35	0.56	0.22	0.70	0.41
The blind	0.70	0.28	0.46	0.36	0.58	0.21	0.71	0.40
**Subjective health status**								
Very good	0.70	0.22	0.36	0.32	0.58	0.19	0.77	0.37
Good	0.69	0.26	0.38	0.34	0.55	0.23	0.60	0.50
Average	0.69	0.28	0.47	0.36	0.57	0.22	0.74	0.33
Poor	0.66	0.30	0.53	0.35	0.59	0.20	0.81	0.25
**Chronic diseases**								
Yes	0.70	0.26	0.50	0.32	0.57	0.21	0.72	0.37
No	0.67	0.28	0.33	0.37 ^h^	0.56	0.22	0.67	0.46
**Other disability than VI**								
Yes	0.69	0.26	0.41	0.36	0.58	0.22	0.70	0.36
No	0.69	0.27	0.45	0.34	0.56	0.22	0.70	0.43
**Quality of life**								
Very poor and poor	0.53	0.33	0.47	0.42	0.55	0.20	0.55	0.42
Average	0.68	0.29	0.48	0.30	0.59	0.22	0.74	0.43
Good	0.68	0.25	0.39	0.35	0.55	0.22	0.68	0.40
Very good	0.79	0.21 ^i^	0.48	0.39	0.58	0.22	0.73	0.35
**Assistant of PwD**								
Often	0.75	0.29	0.54	0.37	0.70	0.19	0.76	0.37
Seldom	0.68	0.28	0.35	0.43	0.55	0.25	0.60	0.49
Never	0.68	0.26	0.45	0.31	0.55	0.20 ^j^	0.73	0.37
**Moving around with help**								
Yes	0.71	0.25	0.44	0.36	0.57	0.22	0.69	0.44
No	0.63	0.29 ^k^	0.42	0.32	0.56	0.22	0.73	0.30

^a^ z = 2.324; *p* = 0.020; d = 0.187; ^b^ z = 2.141; *p* = 0.032; d = 0.190; ^c^ z = 1.971; *p* = 0.049; d = 0.284; ^d^ z = 3.011; *p* = 0.003; d = 0.326; ^e^ z = 2.668; *p* = 0.008; d = 0.254; ^f^ z = 2164; *p* = 0.030; d = 0.285; ^g^ H = 18.237; *p* = 0.0001; ε^2^ = 0.084; ^h^ z = 3.219; *p* = 0.001; d = 0.332; ^i^ H = 8.428; *p* = 0.038; ε^2^ =0.039; ^j^ H = 8.741; *p* = 0.013; ε^2^ =0.040; ^k^ z = 2.120; *p* = 0.034; d = 0.158.

**Table 2 ijerph-19-13519-t002:** Percentage distribution of responses by the dimension of care including the level of disability.

	Total Group *n* = 219				Poor-Sighted *n* = 156	Blind *n* = 63
	Not Important	Rather Not Important	Important	Very Important	Very Important	
**Accessibility**						
I can get appointment easily	0.5	0.5	27.4	71.6	68.6	79.4
Short waiting time on the phone	-	1.4	34.2	64.4	66.0	60.3
I know how to seek health services at night or weekend	0.9	4.6	37.0	57.5	56.4	60.3
PHC clinic is close to where I live or work	1.8	11.4	38.8	48.0	44.9	55.6
PHC clinic has long opening hours	3.7	15.1	47.9	33.3	32.1	36.5
**Continuity**						
PCP has access to prior medical records	0.9	4.6	35.1	59.4	58.3	61.9
PCP is aware of my medical history	-	4.6	51.1	44.3	41.7	50.8
PCP knows about my living situation	17.8	40.6	29.2	12.3	11.5	14.3
**Quality of care**						
**socio-emotional behaviors**						
I understand what the PCP explains	-	0.9	21.9	77.2	73.7	85.7
I feel more comfortable to manage my medical problem after visit	0.4	3.2	47.5	48.9	45.5	57.1
PCP is polite	-	2.7	43.4	53.9	50.6	61.9
People working at reception desk are polite and helpful	-	3.2	32.9	63.9	60.9	71.4
PCP asks about my health problem	-	1.4	37.0	61.6	57.7	71.4
PCP asks about other patients’ problems	0.4	8.2	48.9	42.5	37.8	54.0 ^a^
**preparation for consultation**						
I keep to my appointment	0.9	1.4	38.8	58.9	60.0	57.1
I know which PCP I will see	3.7	5.0	42.9	48.4	48.7	47.6
It is not neccessary to tell a receptionist or nurse about details of my health problem before seeing my doctor	3.2	13.2	47.5	36.0	35.9	36.5
PCP knows my medical background	1.4	9.6	46.6	42.5	39.7	49.2
I can bring my family/friends/assistant to the consultation	11.9	20.5	40.6	27.0	24.4	33.3
I have prepared for the visit, prepared a list of my symptoms and possible questions	4.1	28.3	44.7	22.4	20.7	27.0
**during medical consultation with PCP**						
I do not feel pressure of time	-	3.7	45.2	51.1	49.4	55.6
PCP listens attentively	-	0.5	35.6	63.9	57.7	79.4 ^b^
PCP takes me seriously	0.4	0.9	26.9	71.7	73.7	66.7
PCP understands me well	-	0.5	37.4	62.1	63.5	58.7
PCP is respectful during physical examination and does not interrupt me	-	2.3	32.4	65.3	62.2	73.0
PCP knows when to refer	-	-	24.7	75.3	76.3	73.0
PCP treats me as a person, not just medical problem	-	1.8	37.0	61.2	58.3	68.3
I am honest and do not feel embarassed	-	2.7	41.6	55.7	51.9	65.1
PCP asks if I have understood everything	0.4	7.8	47.5	44.3	44.2	44.4
PCP aks if I have questions	-	8.2	52.5	39.3	37.8	43.9
PCP makes eye contact	14.6	17.8	45.2	22.4	24.4	17.5
PCP asks how I prefer to be treated	2.3	14.6	47.0	36.1	34.6	39.7
PCP avoids disturbance by call, etc.	1.4	10.5	52.5	35.6	33.3	41.3
PCP is not prejudiced (age, gender, religion, culture, disability)	1.4	10.5	36.5	51.6	51.3	52.4
I am open to talk about the use of other treatments	13.7	24.2	41.1	21.0	20.5	22.2
PCP gives me additional info about health problem	9.1	26.0	40.7	24.2	22.4	28.6
I am prepared to ask quesitons and take notes	9.1	29.7	38.4	22.8	21.8	25.4
I tell the PCP what I want to discuss in the consultation	1.3	11.0	58.0	29.7	27.6	34.9
Psychosocial problems can be discussed	23.7	36.1	27.4	12.8	12.2	14.3
PCP informs me about reliable source of information e.g., websites	15.1	29.2	34.2	21.5	16.7	33.3 ^c^
PCP is aware of my personal and socio-cultural background	12.8	26.0	38.8	22.4	22.4	22.2
**After the consultation with PCP**						
I have clear instructions from PCP what to do when things go wrong	0.5	1.4	26.0	72.1	73.1	69.8
I adhere to agreed treatment	-	4.6	41.5	53.9	52.6	57.1
I inform the PCP how the treatment goes	1.4	6.9	47.9	43.8	44.2	42.9
I can see another PCP if I think it is necessary	1.4	12.8	41.6	44.3	44.2	44.4
PCP offers telephone or mail contact in case of further questions	2.3	13.7	50.2	33.8	34.6	31.8
PCP gives me all test results	5.0	6.8	40.6	47.6	50.6	39.7
**Equity**						
PCP involves me in decision making	0.9	5.0	41.6	52.5	52.6	52.4

^a^ Φ = 0.148; ^b^ Φ = 0.204; ^c^ Φ = 0.184.

**Table 3 ijerph-19-13519-t003:** Psychosocial competencies of staff as desired by patients with VIs.

Non-Verbal Aspects of Communication
An eye contact *“I will feel that the doctor is discounting me unless he looks at my eyes.”*
Touch *“I have a very good PCP doctor—when he says “good morning” to me he shakes my hand or touches my shoulder—then I know he is talking to me.”*
Voice messages and their quality *“(…) I remember one appointment I went to with my wife. The doctor did not speak at all until I had to ask my wife if the doctor was in the office. But he was. After that, he started to speak.”* *“[it is important] for the doctor to speak carefully. People with good vision do not realize that eye contact during a conversation plays a big role in understanding what the speaker is saying (…).”* *“I read the other person’s lips so the current situation where the doctor wears a mask at the appointment [in relation to the COVID-19 pandemic] is uncomfortable for me, I don’t often understand, the sound then is also different and I ask the doctor to repeat”(deaf-blind woman).*
Communication through written words—PVIs need: *“PCP to print out all information on a computer in black and white rather than write it down on a piece of paper by hand.”* *“PCP to fully explain the treatment procedure and clearly write (or print) the dosage and names of medications taken during the treatment.”* *“Well done website and detailed description.”*
**Medical staff attitude toward patients with VIs**
Equal treatment by direct communication *“I wish doctors would talk to us—the blind and not ignore us and instead talk about us to the person we came with. When I am at the doctor’s with my son—I can understand it, but if I am with an assistant—I cannot stand it. They treat us as if we are mentally handicapped, as if we have mental limitations. I feel like an invalid patient, it’s embarrassing and humiliating. Assistants do not want to enter the office with us, so that we are not treated in such a way. When we are left alone with the doctors, they have no choice but talk to us. But this is not a solution either. If I have a lot of different documents on me and I do not know which ones I am supposed to hand in and, in which order, I need such a person [an assistant of PwDs]. Doctors should learn how to treat patients who are poor-sighted or blind; after all, they have psychology classes as part of their studies.”*
Serious, empathetic and not indulgent treatment of the patient *“I once heard from a doctor <<Thousands of people live with such myopia>>. It was not pleasant, it took me a long time to come to terms with it and such a message was not helpful.”* *“(…) it is important that a person with visual impairment be treated normally and equally well as other patients. This refers not only to discrimination but excessive attention, signs of pity.”*
Focus on the current patient’s problem and not on the disability *“(…) the doctor should concentrate on the matter with which the patient came, and not ask about the reason why he lost their eyesight and not be disappointed with the patient.”*
**Understanding PVIs limitations and incapabilities related to their disability** *“Doctors don’t know how to behave when dealing with a blind person, they don’t know the problem* e.g., *the doctor shows me the door but it would be better if he walked me to the door.”* *“[It is important] that the physicians know how to ask how to help (avoiding pushing, pulling, etc.). 1. They should be taught this at university; 2. they should be taught that it is not embarrassing not to know, it is embarrassing not to know how to ask.”* *“(…) [important is] the form and language of the doctor e.g., [he says] “let’s see” and he shows me results of the tests and I have to remind him that I cannot see.”* *“Blind people are characterized with greater sensitivity to external stimuli (touch, pain), which is misunderstood and misperceived by medical personnel. There are no procedures to eliminate this misperception.”*

Respondents’ quotes are given in italics.

**Table 4 ijerph-19-13519-t004:** Two aspects of PHC clinic accessibility indicated by PVIs.

Architectural Adaptation of the Clinic Space and Surrounding	Equipment and Technologies
well-marked surgeries, level differences and stairs; unobstructed corridors; decent lighting in the facility; adequate ergonomics of the waiting room—e.g., no chairs in the middle of the corridor and tables with leaflets right under the door; appropriate contrast and font size of documents, Braille lettering above a door handle; voice announcements (activated upon entering the office); a large font on information boards; large numbers and letters on doors (room number, doctor’s name, name of the clinic); a large display of numbers of patients who are to come for an appointment.	“*The ability to bring my own medical equipment with me*.” “(…)* an information system which would enable to identify the location of offices, reception desk or restrooms. This could be, for example, the Totupoint system—it is very useful and widely used. At the same time, it helps a blind or partially sighted person find their way around the space without being assisted by other people*.” “(…) *recommendations printed out on the computer because then the screen reader will read it to me*.” “*It would be important to me if I could record the doctor’s voice and play it back later in home environment*.”

Respondents’ quotes are given in italics.

**Table 5 ijerph-19-13519-t005:** Specific needs of patients with VIs.

**Help with Orientation and Spatial Navigation** e.g., showing the way to the office, from the couch to the chair, walking to the office, walking to the door, helping to go out of the building.
**Help with Queuing** *“At the clinic, I get a number but I can’t see it, someone has to tell me who I’m after, what my number is—so that there is a person who would help.”* *“Being informed by a lady at the reception desk that I can now enter the surgery.”*
**Support with Documents** Support in reading and completing medical records. *“Support with paperwork, arranging, gathering, signing, etc.”* *“Handling documents directly into my hand.”*
**Assisting with any Activities Required for the Examination** Assistance from both clinic staff and an assistant of PwDs *“It is important there be someone in the clinic who knows about my disability and takes care of me there because, for example, in the office the physician only says: “Please sit down and I don’t know where.”* *“Before the visit it is important to introduce in Poland a free assistant of a person with disability who would take me to the clinic, direct me and drive me home.”*

Respondents’ quotes are given in italics.

## Data Availability

The data presented in this study are available upon request from the corresponding author.

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
