# Peer review of "“To Be Treated as a Person and Not as a Disease Entity”—Expectations of People with Visual Impairments towards Primary Healthcare: Results of the Mixed-Method Survey in Poland"

_ijerph, 2022, doi:10.3390/ijerph192013519_

Round 1
Reviewer 1 Report (Previous Reviewer 2)
In the first place, the changes have seemed appropriate to me, as well as the answers to the questions that I raised.
That said, and given the impossibility of grouping the different degrees of visual impairment in more detail, I still miss, at least, a definition that explains the limits of visual impairment and blindness. I think it would help the reader understand the information more properly.
Author Response
Please see the attachment.

This manuscript is a resubmission of an earlier submission. The following is a list of the peer review reports and author responses from that submission.
Round 1
Reviewer 1 Report
This research presents a very interesting proposal and where there is a need for research to improve the health service for blind and myopic patients. In addition to being associated with a valuable European project such as QUALICOPC.
However, I consider that there are aspects that need to be improved:
The objectives are not formulated in the infinitive, but in the form of a question. In my opinion, this form of formulating the objectives is not correct and should be done in the infinitive.
On the other hand, the formulation of hypotheses for each of the objectives would be appreciated.
It is necessary for the authors to indicate whether the QUALICOPC questionnaire has been used in the past in populations with visual impairments (PVIs).
In the description of the sample, it would be very interesting to specify the different VI, between blind and poor-sighted people, as the perception they have through the visual system is different. Poor-sighted people are able to differentiate movements, objects, people, among other aspects of the environment. This fact makes their perception different from that of a blind person. This is why I propose that this aspect should be taken into account in the results section, even being able to differentiate the results on the basis of the visual impairment of the patients.
With regard to the analysis of the open-ended questions, which software was used to analyze the verbalizations? And what statistical analysis was used? As it is presented, it is a descriptive analysis of the answers, but it lacks statistical analysis that really informs about the representativeness of these verbalizations. In my view, this part of the results is not sufficiently relevant because of the type of analysis that has been carried out. I am sure that the results could have more relevance with a proper analysis.
Reviewer 2 Report
First of all, I would like to congratulate you for your work. Reading it has seemed very interesting to me and I value your effort in view of the need for studies focused on people with visual difficulties.
- The introduction seemed appropriate. It adequately contextualizes the subject and is very explanatory regarding the characteristics of the Polish health system.
- Regarding the "Material and methods" section, one of the main drawbacks I find is that the participants self-declared visual impairment, without providing any document that certifies their deficiency. The heterogeneity and broad spectrum that visual impairment and blindness imply mean that grouping all cases under the same "umbrella" of visual impairment, in my opinion, is not the most appropriate. The different degrees of visual impairment make the self-perceived needs differ enormously.
In addition, the period of time in which the data was collected may have affected the responses since, after the recently experienced pandemic, all health systems worldwide have been altered in their normal operation.
On the other hand, the treatment of the data and the aspects analyzed have seemed to me to be exposed in a clear, concise and adequate way.
- The results are clearly exposed and the discussion is also well worked out.
- The conclusions are adequate with respect to the work presented.
Congratulations again for your work.
Reviewer 3 Report
The manuscript written by Binder-Olibrowska et al., focuses on the Poland primary care, exploring the needs of Polish persons with visual impairments when they use primary care services. The study, taking advantage of questionnaire, statistical and content analyses, pinpointed the demand for multilevel interventions, people-centered approach, thus to maximize the chances of meeting different healthcare needs.
The manuscript is well-written, the parameters analysed in the survey are interested and also the analyses performed using it. The use of the detailed tables and the figures help the reader to understand what was assessed. The implication of such study could be helpful for the personnel who works in the field and also to who is approaching to it for the first time.
The manuscript can be published in the present form having addressed these minor points listed below.
Carefully check how to write the paragraph title since they are different (some in bold, some not). Modify accordingly to Journal Author’s guidelines.
Remove revision in the abstract and along the bibliography
Line 22: Please specify what QUALICOPC means.
Line 41: Please modify square brackets into round ones.
Line 50: Please specify what WHO means
Line 61: an “r” is probably missing. Please correct.
Line 66: a reference is missing. Please correct.
Line 71-72: is there a reason why specifying that mostly females aged 18 to 83 years participated in the part of the Polish study regarding patients’ expectations on 219? Were the results of the questionnaire different between females and males?
Line 118: Please specify what PVQ means.
Line 120: no need to specify Supplementary Materials, there is Table S1 already to indicate it.
Please place blank spaces between the table, especially between table 3 and 4, and check the font of the Table titles.
